# Household Food Security as a Complex System—Contributions to the Social Sciences from the Cuban Perspective during a Pandemic

Yinet Domínguez Ruiz [1,*], Osmanys Soler Nariño [1], José Manuel Jurado Almonte [2,*] and Juan Antonio Márquez Domínguez [2]

1 Departamento de Sociología, Facultad de Ciencias Sociales, Universidad de Oriente, Santiago de Cuba 90500, Cuba
2 GI Instituto de Desarrollo Local, Departamento de Historia, Geografía y Antropología, Universidad de Huelva, 21071 Huelva, Spain
* Correspondence: yinetd@uo.edu.cu (Y.D.R.); jurado@uhu.es (J.M.J.A.)

**Abstract:** The food crisis caused by the rise in grocery prices affects many countries. Added to this complex panorama is the current health situation generated by the COVID-19 pandemic. Its impact on the productive sector will be detrimental to many household incomes and food practices. The social sciences need to adopt a complex understanding of household food security (HFS) as a dynamic process of building collective nutritional knowledge and eating habits. In the case of Cuba, the burden on the agrifood system is the result of external and internal factors that affect household food sustainability. This paper, therefore, seeks to assess the social construction of HFS as a complex system in the current pandemic scenario using the municipality of Santiago de Cuba as a case study. Qualitative and quantitative methods were used. The main results obtained focus on the role played by women in food use and distribution, and the effect of food vulnerability on HFS. These results provide an analytical model for the study of the new and diverse household-food-security configurations that are emerging.

**Keywords:** household food security; social sciences; COVID-19; Cuba

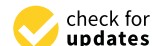



## 1. Introduction

Eating is an action of which the symbolic construction is conditioned by social, economic, cultural and political factors that stem from the complex relationships between norms, values, practices and actors. At the household level, these factors produce and replicate vulnerabilities and inequalities in the everyday access to food. In the case of Latin America and the Caribbean, this access has been marked by the decline in income from employment—a situation that is associated with work restrictions, as well as the limited possibilities of remote working within the current pandemic context.

The economic indicator is globally connected to the sociopolitical structures of each society, where the disparity in the distribution of resources is currently exacerbated by the COVID-19 situation. In this regard, by 2020, this health crisis had generated further food destructuring in many households in Latin America and the Caribbean because of the rapid emergence of social factors such as unemployment and vulnerability, increasing poverty by 34.7%, compared with 30.3% in 2019 [1].

Within this scenario, gender inequalities in the home were also exacerbated because women's unpaid household chores increased with lockdown measures, as they had to pay greater attention to children, the elderly and people with disabilities, among others [2]. About 80% of unpaid care work is carried out by women in Latin America [3], which highlights the impact of certain social and cultural factors on family dynamics. This includes the sociocultural replication of gender representations, stereotypes or stigmas.

The Economic Commission for Latin America and the Caribbean (ECLAC) (2020) [2] stresses the consequences of the pandemic on the female sector by highlighting the incorporation of women into the informal labor market (77.5% of the total). This insertion is developing under vulnerable and precarious conditions, aggravating the possibilities of purchasing assets and resources to financially protect the groups under care.

Many of the problems mentioned are the result of the complex relationships and interactions established by actors and factors in the social construction of household food security (HFS). As a complex system, this security is structured by the action of public policies and the treatment of food-insecure families. The latter has connotations for the 2030 Agenda and its SDGs, and specifically Goal 2, the priority of which is to achieve greater food security to guarantee the social well-being of all people.

For the social sciences, the field of household food security has had limited theoretical influences [4–11]. Although the referenced authors use analytical frameworks close to this area of scientific knowledge, the study of the interrelationship between the dimensions of food security (access, use, stability and availability), as well as the study of the factors (social, cultural and economic) that integrate or disintegrate that security, is still insufficient.

These epistemic gaps are also reflected in the research carried out by international organizations, such as the Food and Agriculture Organization of the United Nations [12,13], the Economic Commission for Latin America and the Caribbean (ECLAC) [1,2,14] and the Community of Latin American and Caribbean States (CELAC) [12], among others. On the one hand, most of them focus on food access, utilization, stability and availability, and, on the other hand, on the factors that condition food security and situations of social disintegration, which are exacerbated in the current pandemic context.

All of the above reflect the urgency and importance of the aforementioned subject matter, which recognizes the challenges posed by COVID-19 for the social sciences in relation to the study of HFS as a complex system. Achieving participatory, sustainable, inclusive and accessible food security for all is a challenge given the current economic crisis that is exerting a negative effect on the availability and stability of food, which paints a bleak social and food picture. Hence, the social sciences need to adopt new theoretical and methodological approaches that incorporate the complex conception of HFS as a dynamic process of collective construction. Its application in social reality is influenced by concepts, variables and dimensions that are conditioned by the public policies designed by each country to meet individual and collective food needs.

In Cuba, the social construction of this food security is the result of the combination of the external factors (economic and commercial blockades) and internal factors (insufficient interconnection between the production, distribution and marketing of food) that have a marked impact on the food sustainability of households. Given this situation, and in the midst of a complex health scenario, in July 2020, the Cuban state passed a policy to promote its territorial development, as well as the Food Sovereignty and Nutritional Education Plan. The first seeks to foster local development; the second is geared towards strengthening food sovereignty in complicated national and international economic scenarios.

At a local scale, recent studies [15–18] have analyzed how family conflicts and social, environmental and structural vulnerabilities affect access to food. Through a diagnosis carried out in two communities (Chicharrones and Los Maceos in the municipality of Santiago de Cuba, 2021), the aforementioned problems revealed the constant fragmentation between the constitutive elements of food security (actors, factors and dimensions of analysis) in households.

This paper therefore seeks to assess the social construction of household food security as a complex system in the context of the pandemic in the municipality of Santiago de Cuba in order to develop actions that boost public food policy at the municipal level. For this purpose, the general methods of scientific knowledge were applied, as well as qualitative and quantitative methodological perspectives. Of the latter, methods and techniques such as scientific observational studies, interviews and surveys were used. Data

triangulation was also applied to validate the results obtained on the issues identified in the two communities under study (Chicharrones and Los Maceos).

This study aims to introduce a perspective of HFS as a complex system in which different actors (institutions, organizations and households) and factors (social, economic, health and cultural) interrelate in the food space. This contribution incorporates an analytical model for the study of family relationships (food, economic and intergenerational, among others) and configurations (access, use, stability, availability and gender) that emerge in said food security. All are affected by the COVID-19 pandemic.

## 2. Theoretical Framework

### 2.1. Theoretical Considerations on Household Food Security in the Framework of Social Sciences

Household food security has anthropological roots that play a role in attitudes towards making, experiencing and understanding food in the family–society–cultural-system relationship. This vision shows that, in the field of household food security, not only financial, biological and legal elements converge, but also cultural structures that symbolically mark social relations [19,20].

Unlike anthropological studies, sociological research in the field of food points to problems in consumption, food-use behavior, food destructuring and food systems, to name but a few. In this regard, several authors highlight the delay this science has made in interpreting the food phenomenon along three specific lines: (1) the concept of food as a biological and daily necessity; (2) the limited theoretical approaches to food consumption; (3) the absence of value judgements on the priority of hunger as a social issue in developed Western societies [19,21–23].

The current sociological research addresses three theoretical debates that have an impact on the social treatment of food [11,24–26]. The first refers to the degree of stability or destructuring of contemporary food. It focuses on the processes, practices, habits and behaviors that stabilize or destructure nutrition in individuals, groups, families and institutions. A second debate concerns the social-class factor as a generator of food norms. From this analytical perspective, the position that groups and individuals hold in the social structure determines their behaviors, lifestyles and food norms.

A third debate addresses the need to establish (or reset) the link between production and consumption. Today, food industries manufacture products that are consumed by a large part of the population that, in turn, have unforeseen health consequences. Thus the importance of integrating production and consumption processes in order to enhance sustainable ecological and inclusive food security that meets the dietary needs and preferences of all social groups. However, while the latter position points to the relationship between production and consumption, there is still an insufficient understanding of the different dimensions of food security within the frameworks of this relationship. For this reason, studies need to delve deeper, as it involves actors, factors and dimensions in constant interrelation not only at the macro level, but also at the microsocial level.

With regard to the concept of food security, much research remains to be conducted in the social sciences. Although its definitions have been diverse, one of the most significant definitions for this area of scientific knowledge was given at the World Food Summit of 1996, where the statement was coined that food security exists when: "All people, at all times, have physical and economic access to sufficient, safe and nutritious food to meet their dietary need and food preferences for an active and healthy lifestyle". This definition indicates that food security has four main dimensions: (1) physical availability, (2) economic and physical access, (3) utilization, and (4) stability over time.

As we see it, the above definition introduces important contributions. The first is based on the incorporation of the dimensions of the analysis of food security. A second contribution lies in the possibility that these dimensions offer to substantiate the concept in a more practical way. The third part focuses on studying these components in terms of factors that may have an impact on the main problems of access to food—for example,

climate change, the consequences of incorrect eating behaviors or the financial factor, given that food prices are high and incomes are low, to name but a few.

However, there is insufficient coordination not only between these components, but also between the different actors (individual and collective) and factors (social, economic, health, political and cultural) that contribute to the social construction of HFS as a complex system. Another reduction lies in the limited treatment of the role of the family as a social actor in the food security of its members, based on its reciprocal relations with these social actors (organizations, groups, families and institutions).

Despite these theoretical weaknesses, the first half of the 1980s marked the establishment of a concept of household food security [27], the essence of which lies in the following: access by all people, at all times, to sufficient amounts of food for an active, healthy life. Its essential elements are the availability of food and the possibility of purchasing it. This position marked a research turning point in the treatment of household food security. However, this concept is still in a macro position because it reduces important elements of the family way of life, such as food practices, habits, norms and representations.

This household-focused approach to food security has opened up new theoretical opportunities within the social sciences [5–8,10,28–30]. Its fundamental contributions continue to be framed within the dimensions of food access, use, stability and availability, without illustrating the relationship that these maintain with each other. This reflects an insufficient level of research into the relationships between these components, and the social, cultural and economic factors involved in the integration or disintegration of food at the household level.

In Cuba, studies on HFS have focused mainly on the area of social food policies and the factors that condition it [31–35]. In general, these studies show the importance of this topic in terms of promoting more sustainable and inclusive food systems. However, distancing from the practices, interactions and meanings constructed around the use and consumption of food is still evident, which limits the understanding of HFS as a complex system.

*2.2. Household Food Security as a Complex System: Its Social Construction from the Links between Actors, Factors and Dimensions*

The household is constituted as a constantly changing system of rules, norms, symbols and meanings [36]. It is shaped by the dynamic interplay of its members, which thereby defines the life cycle of each household. Likewise, the family is a symbolic system, the complexity of which is structured or organized by the confluence of elements such as proximity, emotions, affections, behavioral patterns and norms. They are an integral part of the different subsystems of the family (marital, parental, and sibling) and are involved in the biological, economic and educational functions of the family.

For many authors [37–42], this family system is influenced by a number of associated variables—for example, family structuring in terms of family typology, domestic violence, gender, care, and social, economic and cultural vulnerability. They all affect the functioning of the family, generating conflicts, integrations or deviations in the behavior that social subjects develop within or outside this system. However, while this research addresses important issues for sociological family studies, there are still not enough insights into the role of households in food security.

The issue of food in the studies cited above is downplayed in relation to other variables, such as gender, family life strategies, family functioning and socialization. Several studies [4,20,43–49], more focused on the relationship between households and food, address eating habits or behaviors, eating disorders, the role of women in household food security, food consumption in situations of poverty and the social, economic and cultural changes that affect family nutrition.

Nevertheless, most of these theoretical stances fail to treat the family as a system of complex social practices and interactions around food. Their contributions are fragmented and insufficient to understand the various symbolic exchanges produced between the dimensions of HFS, the actors and factors that condition food destructuring.

We believe that to achieve this family–food relationship or interdependence, new analytical positions need to be introduced that establish household food security as a complex system. This theoretical perspective is based on Luhmann's theory [50] on autopoietic systems. Such systems are closed by a recursive circularity, but, at the same time, they are open systems because their self-reference implies relations with themselves and with the environment. This self-reference is aimed at differentiating from this social context, which is made up of other systems: family, economic, political and cultural.

From this theoretical perspective, two important concepts of this complex system should be added: the structural connections and operational provision. One is found in the capacity of the different structures within the system to interrelate in the face of the complexity of the environment; the other is framed by the way in which these systems are sealed off from it, enabling it to organize its structures and operations to fit into the social environment [50].

In the case of HFS, it is constituted as a complex or autopoietic system that is based on, first, the ability to continuously self-replicate through its own structures (norms, values, meanings and practices, among others). Second, it develops its main elements similar to any other social system; for example: (1) economic, biological and educational functions; (2) communication between family members; (3) establishing priorities for food access and distribution; (4) food practices in relation to processing and consumption; (5) the socialization of individuals and attention to vulnerable groups in terms of food.

As can be seen, this perspective assesses the connections, interactions and feedbacks between the abovementioned elements in the social construction of HFS. However, all of the above are influenced by the action of the symbolically generalized media or symbolic mediations. These means or structures are power (or power/right), scientific truth, money (or property), love, art and values [50].

Symbolic mediations also play a role in the conception of a complex system, as they have the capacity to interconnect actors (individual and collective), factors and the dimensions of HFS (food access, availability, stability and utilization). For the purposes of our research, these mediations are power (gender perspective and food distribution), money (economic income, physical and economic access to food at the household level), language, norms, meanings and symbols.

When referring to power, this is reflected in the unequal roles assumed by women in the distribution and preparation of food, as well as in the care of vulnerable groups. This reflects an androcentric culture constituted by norms, discourses and practices that legitimize gender inequalities, where women occupy disadvantaged positions in terms of the overload of domestic chores.

Money and income allocated to food purchases is not far from this symbolic power. The latter is marked by gender gaps between men and women regarding care for vulnerable groups. Generally speaking, many of these financial difficulties are linked to the low income and purchasing power of family members to achieve subsistence and maintenance.

This leads to various practices that are produced or replicated in the family in order to overcome these food contingencies. Some are aimed at solving the basic needs of their members (stocking, controlling or planning food); others generate conflicts in terms of accessibility and use (organization of the family budget in relation to fluctuations in market prices). Although both practices (integration and conflict) characterize the complex system, their production or replication is the result of the incidence of social, economic and cultural factors that structure the family way of life.

These factors generate conflicts within the family through the mediation of language as a symbolic structure. Through communication (language) consensus, conflicts, agreements and cooperative practices are achieved that enable social actors (family, institutions and organizations) to address the problems generated.

Within food processes, social actors are involved in the construction of meanings, beliefs and knowledge around household food security. These construct cognitive structures

or thought schemas that are incorporated through the process of socialization in different social contexts [51,52].

For this paper, the social actors fall into the following categories: family, community organizations and institutions. Their connection with the factors (social, economic and cultural) constitutes an important element in the social construction of family food security as a complex system. Therefore, such a system highlights the relationship established by these actors to solve the demands of society, as well as to design opportunities for greater access to social food policies, according to the needs and potentials of families.

In short, the proposed perspective of food security is summarized as a process of physical, economic and social access to sufficient nutritious and safe food, in which various factors and social actors are interrelated. These elements are dynamically linked in the household–environment relationship through symbolic mediations that generate processes of change, conflict, integration or food destructuring. These processes map the satisfaction of food needs and preferences onto the linkages and interdependencies established by the dimensions of food access, stability, use and availability. In this regard, its self-organization contributes reciprocally between actors and factors to the social construction of HFS as a complex system.

### 2.3. Approaches to Food Security in Cuba: Gaps in Understanding Household Food Security as a Complex System

The food-system model that prevailed in Cuba until 1989 sought to guarantee the availability of food in terms of the sufficiency and stability of the basic supply. In practice, these objectives were achieved independently of the productive performance of the agricultural sector. Access was also ensured through the rationed and social consumption of products at state-subsidized prices, which underlined the principle of equality in food distribution [53].

The crisis of the 1990s led the Cuban political system to transform in order to meet the needs of the various sectors of the population, and especially the most socially disadvantaged. An example of this is the promotion of the Food and Nutrition Security (SAN) program to ensure access to food for the most vulnerable groups. Conflicts during that decade resulted in the emergence of contradictions around food production, institutional control, the stability of distribution networks and high market prices for the production of goods and services [54].

This situation led to the approval of the National Action Plan (PAN) for nutrition in 1994, and its subsequent ratification in the Report of the Republic of Cuba presented at the World Food Summit in 2002. Currently, the main policy framework for the public policy on food security in Cuba until 2030 is the Economic and Social Development Plan, with its strategic axes and sectors. It highlights the axis of "Human development, equity and social justice" that is aimed at, among other elements, preserving not only universal policies (health, education, social security and assistance, culture, among things), but also domestic development in terms of food access, stability and availability.

With the emergence in 2020 of the COVID-19 pandemic, the inadequacies in Cuba with respect to the relationship between production, distribution, marketing and consumption led to the incorporation of new concepts to promote national productive development. As a result, a number of measures were taken to achieve secure access to food for the groups most vulnerable to the pandemic. The measures include protecting those susceptible to infection, as well as delivering food parcels at the community and workplace levels to ensure that food distribution can reach everyone.

Despite the measures applied in the face of HFS, the effects of COVID-19 revealed that a solid food-security policy in this country depends on the interplay of various factors at the macro- and micro levels. On the one hand, at the macro structural level, the financial constraints of the international context, global imbalances in economic relations and social inequalities between countries come to the fore. On the other hand, there are micro contradictions, such as insufficiencies in the distribution networks of goods and services

due to limitations in the sustainability of resources that make it possible to meet the food preferences of individuals and social groups. In addition, there are problems related to the organization of public and private spaces, the feedback of social actors, the diversity of offers and the high cost of household consumption.

All of the above reveal some shortcomings in understanding household food security as a complex system from the Cuban reality. In a nutshell, the most important are:

- Reduction in the effects on the interactions, meanings and daily practices of households regarding food access, use and consumption in the microsocial environment;
- Insufficient treatment of the complex processes (consumption, distribution, production and others) established in the family framework based on social relations around food;
- Reduction in family vulnerability to economic, environmental, physical, geographic and nutritional factors, while ignoring other elements, such as the practices that destructure food and produce and, in turn, multiple forms of dismantling family feeding habits.

The shortcomings highlighted above show that HFS is not limited to food access, use, stability and availability, but includes other important variables (intergenerational, gender, symbolic, among others) in the system of relationships that take place around these dimensions. These variables constitute a sui generis reality of the food phenomenon and its derivation into a complex system. Therefore, our analysis requires a methodological strategy that makes it possible to collect data on these variables in the household context in a relational and integrated way.

### 3. Methodological Strategy for the Study of Household Food Security as a Complex System

The methodological strategy that supports this research is divided into two phases, with methods and techniques for collecting information to analyze, contrast and verify the results. These methods operate at both the theoretical and empirical levels. In general, the empirical ones respond to the qualitative and quantitative methodologies, the combination of which made it possible to obtain different points of view on the results of the study carried out. As for the theoretical ones, they constitute the procedures of scientific knowledge (for example: analysis and synthesis) that have facilitated the examination of various bibliographic sources that refer to the treatment of food security at the household level.

Based on the above elements, we present the main characteristics of these phases:

1. First phase: This is exploratory in nature and was developed with the aim of diagnosing (from the HFS) the main social problems of the Chicharrones and Los Maceos communities in the municipality of Santiago de Cuba (Figure 1). Scientific observational studies and surveys were used at this stage. The observation variant used was the semi-structured model, which provided information on household difficulties in terms of food access, availability, utilization and stability. In the case of the survey, 200 questionnaires were applied, and its objective was to collect standardized information on the situation in the families of the communities analyzed, based on the dimensions of HFS.

2. Second stage: Twenty in-depth interviews were conducted with families from both communities. This technique made it possible to identify the significance attributed by the family to the different social, cultural, health, political and economic factors that shape household food security. A purposive (nonprobability) sample was used to select these households [55], the fundamental criterion of which was based on the fact that the chosen families were in unfavorable situations with regard to food access, availability and utilization.

Fourteen key informant interviews were also conducted, including with the woman presidents of the neighborhood councils, family doctors and the presidents of the Federation of Cuban Women (FMC) and the Committees for the Defense of the Revolution (CDR). This technique sought to identify the subjective perceptions of these interviewees in relation to the constituent elements of HFS—such as the role played by different social groups

and organizations concerning this security, the infrastructure in place, the quality of the services provided in the gastronomic facilities and the average household income, among other elements.

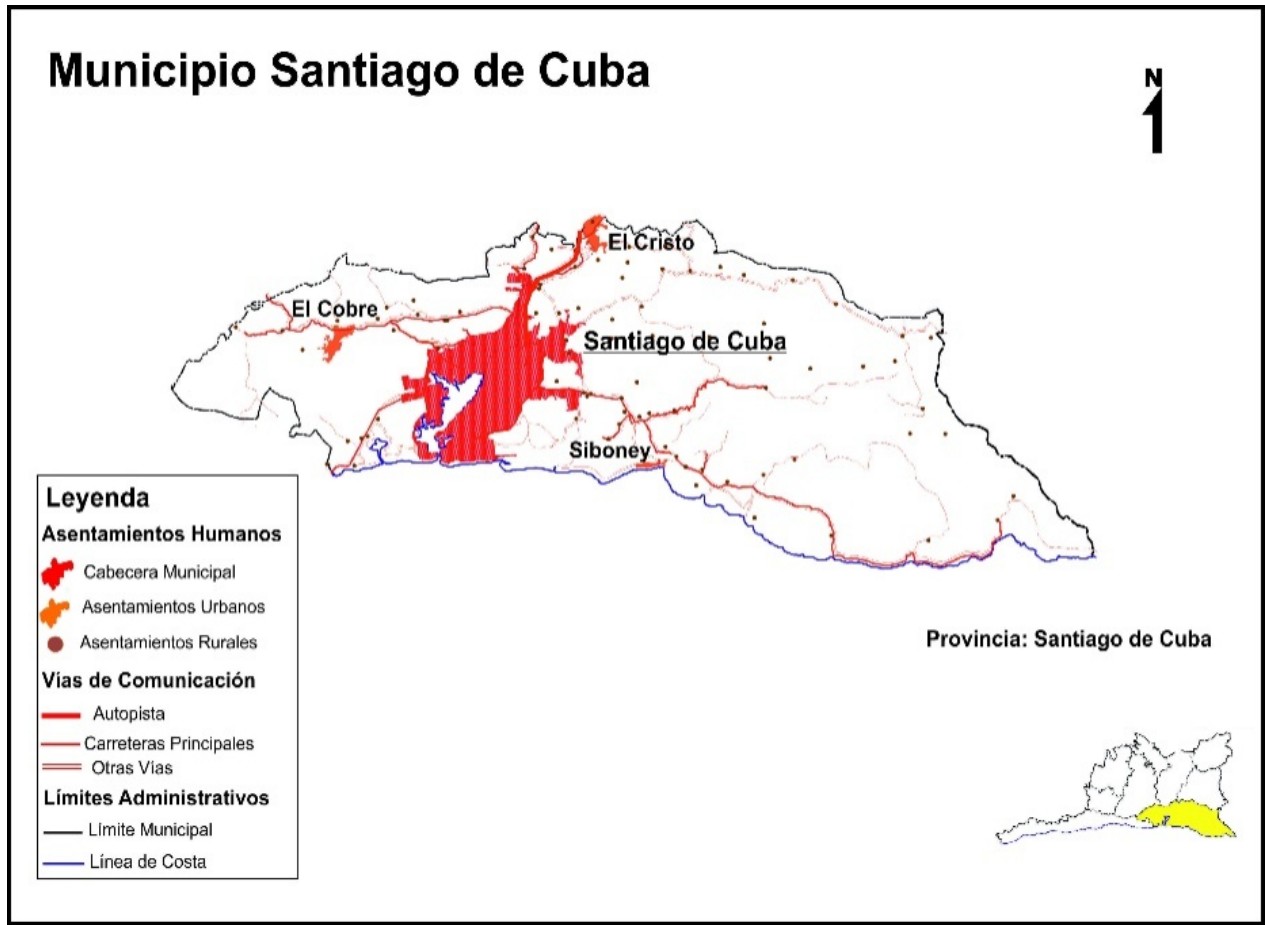

**Figure 1.** Geographical location of the studied communities. Municipality of Santiago de Cuba. Source: Google maps (April 2022).

The interviews were conducted with three experts [56], and specifically on the issue of household food security as a social phenomenon. All experts were selected on the basis of their experience in the field both at the national and international levels. According to Ruiz [55], purposive sampling (nonprobability) permits the selection of experts from the chosen sample by means of a strategic criterion that prioritizes knowledge of the problem being researched.

The survey was applied to a sample of 200 families from a population of 21,631 inhabitants in Chicharrones, and 24,420 in Los Maceos. The objective of this technique was to examine, in depth, the configuration of the different dimensions (food access, stability, utilization and availability) at the household level by integrating different factors and social actors—such as household food practices, community organizations, social institutions and knowledge built about household food security.

This procedure was performed based on a sampling error of 10% and a confidence level of 95.57%. This choice was made on the basis of the characteristics of probability sampling, and in particular, random sampling, in which its fundamental condition is expressed in the idea that all individuals in the population have the same probability of being selected in the sample chosen to constitute the elements of the sample [57].

Finally, methodological triangulation was used to obtain empirical information from different sources and collection methods, and to contrast them with each other [58]. This methodological triangulation constitutes a combination of methodologies in the study of

the same phenomenon [59]. The possibility of incorporating different methods and techniques through triangulation helps to reinforce the relationship between the construction of meanings and the quantification of the households' level of knowledge about food security.

The use of this combination of sources and methods in the sociological understanding of social, cultural, health, economic and political factors made it possible to reach the results of this research from different methodological angles, enabling the interpretation, understanding and explanation of the social reality in relation to food security in the homes of the communities studied.

## 4. Findings in the Study of Household Food Security as a Complex System at the Community Level

The most important results of the research on the social construction of HFS as a complex system in the communities studied (Chicharrones and Los Maceos) of the Santiago de Cuba municipality are presented below. Some of the sociodemographic data of the selected sample show a trend towards an aging population, which is reflected in the age ranges of the people surveyed (Figure 2).

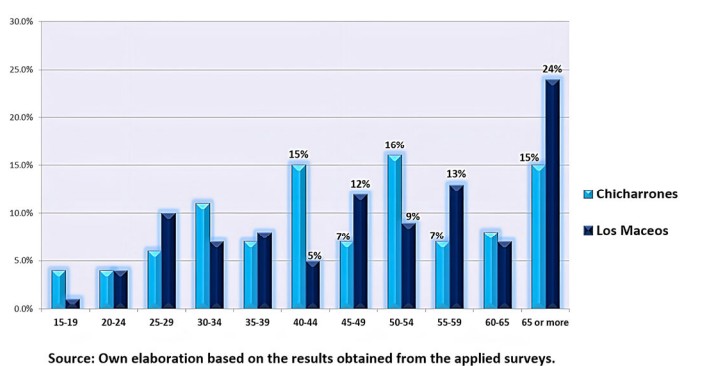
Source: Own elaboration based on the results obtained from the applied surveys.

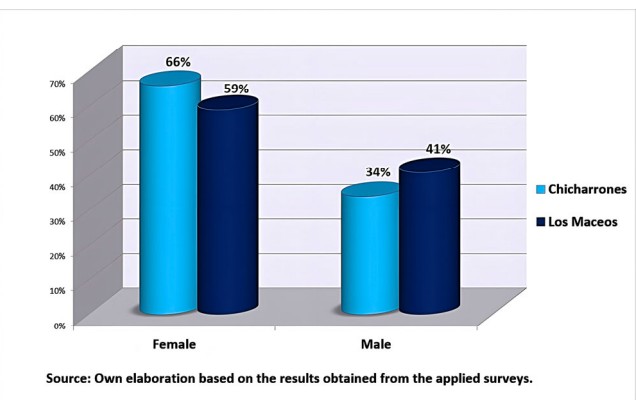
Source: Own elaboration based on the results obtained from the applied surveys.

**Figure 2.** Age ranges and genders of the respondents. Source: surveys (2021) and authors' own research.

In terms of household food security, greater family participation has been required in feeding the elderly in the context of COVID-19. The latter has marked deep gender gaps in the distribution of the household chores and care practices that arise from caring for this vulnerable group. There is a high percentage in the communities studied, which shows a greater involvement of the communities in the achievement of the HFS dimensions.

Regarding these dimensions, women play a significant role in the preparation and distribution of household food (Figure 3). This shows that there are still gender stereotypes that mark women as responsible for domestic chores. This social factor has an impact on family relations, as it is constituted, on the one hand, on the basis of inequalities around food, and, on the other hand, as a mediating structure, the symbolic power of which legitimizes the practices assumed by women in the family sphere.

The people interviewed assert that women still occupy disadvantaged positions in relation to gender roles, not only in the distribution of food, but also in the sharing of household chores. Likewise, their position in HFS is shaped in relation to the values and norms produced or replicated in food preparation and consumption. These include preparation and distribution (Chicharrones: 85%; Los Maceos: 77%), as well as washing food before cooking (Chicharrones: 75%; Los Maceos: 80%).

Regarding the perception of the community's social vulnerability, most reported that this is the result of a number of factors, including financial (weaknesses in the stability and distribution of food) and social factors (population aging) (Figure 4).

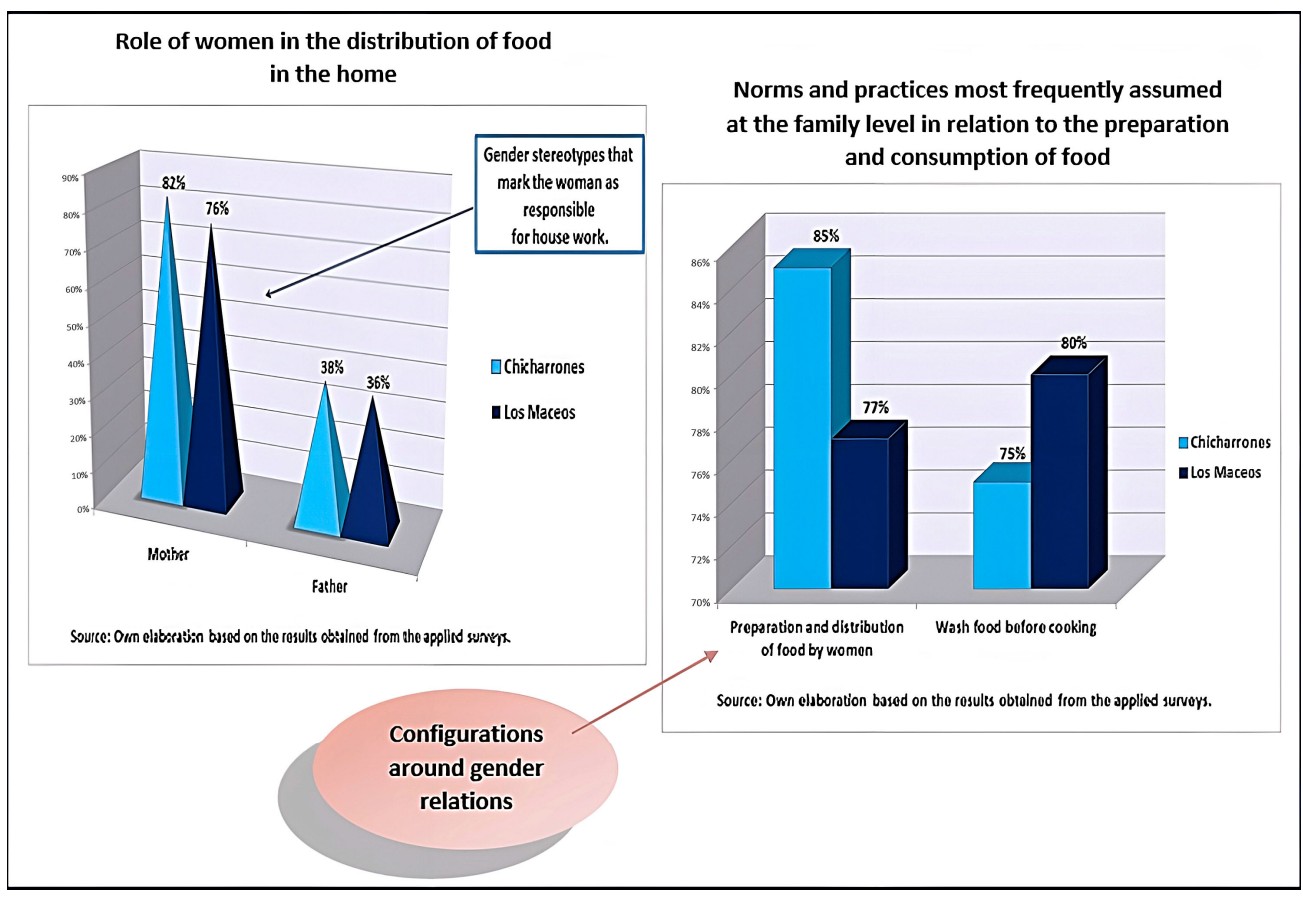

**Figure 3.** Role, norms and practices assumed by women in HFS. Source: surveys (2021) and authors' own research.

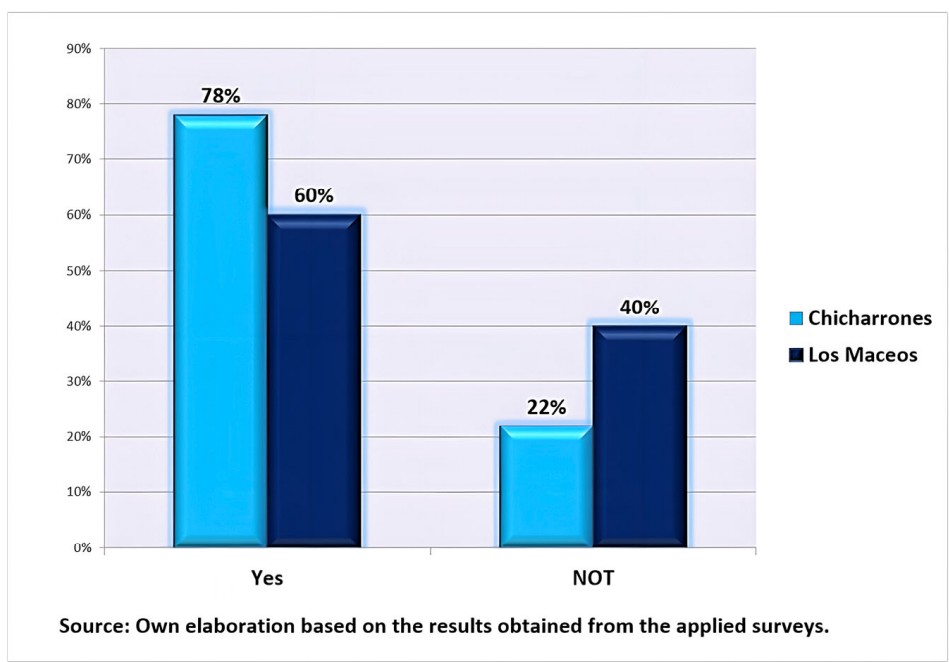

**Figure 4.** Community vulnerability as perceived by social actors. Source: surveys (2021) and authors' own research.

It should be noted that the incidence of this vulnerability in HFS is contrasted with scientific observation, as many households generate inadequate habits regarding the access

and use of the goods necessary for their nutritional needs (Figure 5). In some cases, this stems from cultural patterns that are internalized in socialization processes marked by infrastructural problems, reductions in the food supply, the lack of a socially acceptable food system and overcrowding, among other situations. In this regard, experts agree that these problems are exacerbated by economic reductions arising from COVID-19, which have had a significant impact on household feeding practices.

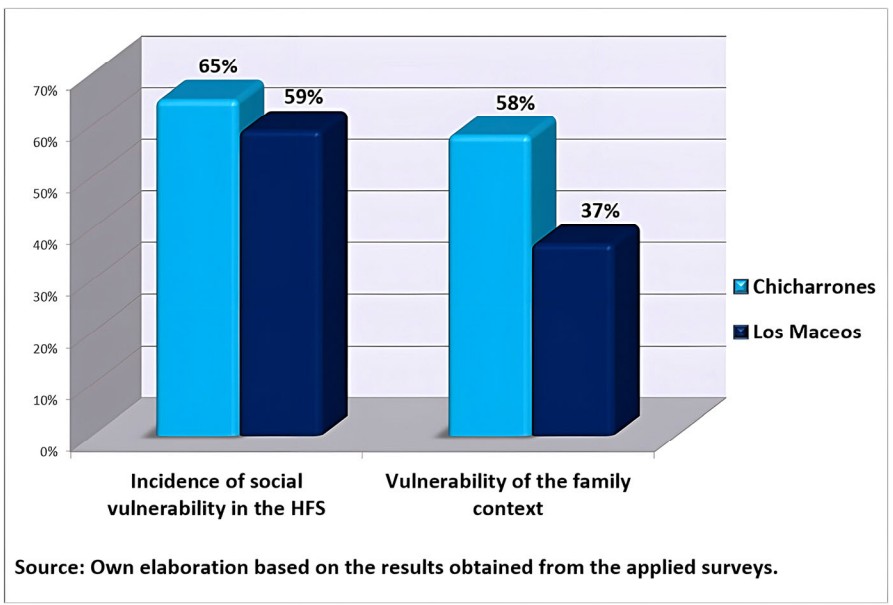

Source: Own elaboration based on the results obtained from the applied surveys.

**Figure 5.** Assessment of the community on the incidence of social vulnerability in HFS. Source: surveys (2021) and authors' own research.

In this pandemic, HFS has been built as a complex system as a result of the interaction of these factors and actors (individual and collective) in the food sector. Its action on the use, access, stability and availability processes conditions the emergence of social problems on a local scale. Specifically, for example, this is the case of fragmentation in the production–marketing–price–consumption relationship, which results in limited household access to the food opportunity structure, depending on the assets or resources they possess or can mobilize.

In our view, the situation described above requires greater communication between social actors (institutions, organizations, markets and families) in these difficult times of pandemic. As a mediating structure, communication enables a dynamic link between these actors based on the exchange of information about needs, interests, preferences and social alternatives to fill the gaps in goods and services at the community level.

## 5. Discussion of Findings and Action Proposals

### 5.1. Family Structures in Terms of Food Security:Municipality of Santiago de Cuba

We assess two lines of explanation that summarize the theoretical and methodological bases of this social construction of HFS as a complex system. These are contributions to the interpretation of the food phenomenon from the social sciences. First, the emergence of configurations is based on the reciprocal relations established between the different actors and factors. Second, each of these configurations is explained in terms of the systemic family relationships and their interconnection with the dimensions of HFS.

To explain the above, the different elements that make up HFS as a complex system in the current context of the COVID-19 pandemic are described below (Figure 6). These elements are the following: the systemic family relationships produced in the homes, and the individual or collective actors and the social, economic, political and cultural factors that mediate these relationships and interactions. The connections between them generate various social constructions that represent new ways of explaining household food security.

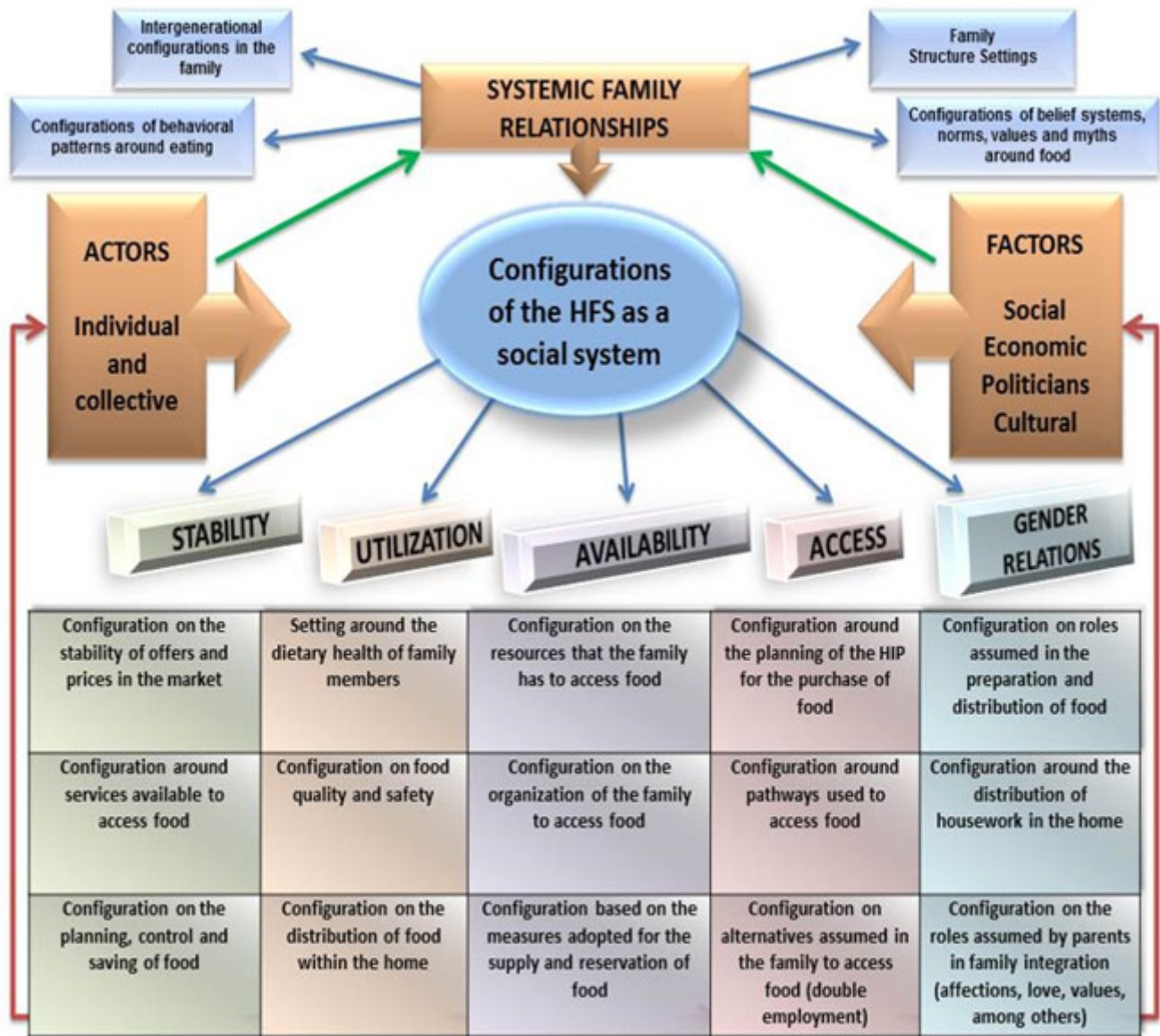

Legend: → The different configurations that emerge from the relationships between different actors and factors. → The reciprocal relations between the actors (individual and collective) and factors (social, economic, political and cultural). → The emergence of systemic family relationships (configurations) from the interconnections produced between actors and factors. Source: Survey results (2021) of the investigation.

**Figure 6.** Configurations that emerge from HFS as a complex system in times of COVID-19. Source: surveys (2021) and authors' own research.

Specifically, the household system reproduces a set of social practices that express the family's way of life at the individual and collective levels. These replicated practices are the result of the complex relationships established not only within the household, but also with the environment. These relationships are connected by various links with underlying norms, rules of behavior, habits, symbols and meanings. Within this process, communication plays an important role in that it allows the individual actors (mother, father, children, president of the people's council, CDR (Committees for the Defense of the Revolution), among others) and collectives (the family, community organizations and social institutions) to interrelate through symbolic mediations, such as language, money or power.

In this interrelation, it is worth highlighting the socialization of these actors for the development of HFS as an autopoietic system [50–52]. However, we consider the necessary complementarity not only between the aforementioned actors, but also their link with the various factors that intervene in the self-organization and coupling of said security as a complex system. Hence, the analysis of the configurations that emerge in this system becomes a superior link in theoretical matters based on the complexity of its study.

All this without failing to recognize that the interconnections between the aforementioned actors do not always occur around symmetrical relationships. Conflicts also emerge that affect food stability, availability and secure access. The presence of an unstable distribution network for goods and services, as well as irregularities in the stability of food to guarantee social institutions, show some of these interaction conflicts in COVID-19 times. Here, it is significant to pinpoint the different problems generated in families as a result of these inconsistencies in the dimensions of food security.

These situations include fluctuations in food prices that bring about changes in household income planning (HIP) conditioned by lockdowns, as well as insufficient actions carried out at the local and community level to achieve greater sustainability in food sales and distribution.

Generally speaking, both problems are the result of the interplay of social, economic, health, political and cultural factors. On the one hand, the macro level includes the infrastructure for the sale of food and public services, and the regulatory or integration mechanisms exercised by institutions to organize the distribution of and access to food. On the other hand, at the micro level, the conditions of social vulnerability, gender inequalities in food care, customs, way of life and the knowledge constructed about food in the family space are highlighted.

The dynamic confluence of these factors and actors in HFS is expressed through systemic family relationships as a social product that emerges from these interactions. These systemic relationships are social, cultural or symbolic processes socialized within the household in terms of food depending on the historical and sociocultural trajectory. These processes have materialized in various configurations dominated by intergenerational links, the family structure, communication and the food habits internalized by the different social actors. These configurations are related to the analysis dimensions of HFS.

Based on the above approach, these configurations are structured as follows:

1.  First: Food stability, which is expressed in the planning and control capacity of families within the health crisis, in which the market-price stability and services available for access are affected. In this regard, systemic household relationships concern the actions that households adopt in order to maintain a balance of assets or economic resources to securely access a sufficient amount of food;

2.  Second: Intergenerational links and the diversity of meanings attributed to food. This reflects practices or configurations associated with food use and consumption. Here, the configurations show, from a relational point of view, the capacity of families to conserve, prepare and distribute food in pandemic conditions, prioritizing attention to vulnerable groups (children, the elderly, pregnant women and people with disabilities), and especially their nutritional needs;

3.  Third: Food availability based on the resources that allow families to have secure access to a sufficient amount of food. This requires cooperation among all members to ensure that responsibilities are shared, and resources are planned. Here, a family structure is formed, linked not only to the functions of the family (biological, economic and educational), but also to the emotional links that result from the relationships between the different subsystems (marital, parental and sibling). Within this structure, the adoption of food-supply and stockpiling measures to cope with the effects of the current pandemic situation is a key element;

4.  Fourth: Access to a sufficient amount of food, the materialization of which depends, to a large extent, on the channels used by families and the alternatives taken to boost income (two jobs). Consequently, pathways and alternatives have strong links

with the planning of the HIP, as, on the one hand, organizing this revenue solves the structural weaknesses present in these pathways, and, on the other hand, the alternatives constitute a significant source of income in order to meet the food needs of family members;

5.  Fifth: Gender relations of women's roles in food preparation and distribution in the household. These are androcentric behavioral positions within the marital and parental subsystem. These equity gaps are structured around myths and beliefs about the social position of women in the family structure. In this regard, for example, women are stereotyped as the people responsible for the household food, whose subjective capacity makes it possible to integrate all the members through emotional ties. However, although women carry out these domestic chores, it should not prevent men from also contributing to family food security in general, and especially during lockdowns and reduced social mobility.

### 5.2. Proposal for Actions to Improve Household Food Security

Below, an action plan is proposed, the general purpose of which is to strengthen the level of HFS in the communities of Chicharrones and Los Maceos in the municipality of Santiago de Cuba. It will draw on those configurations to enhance the complex dimension by interrelating food access, use, availability and stability at the household level during a pandemic (Figure 7).

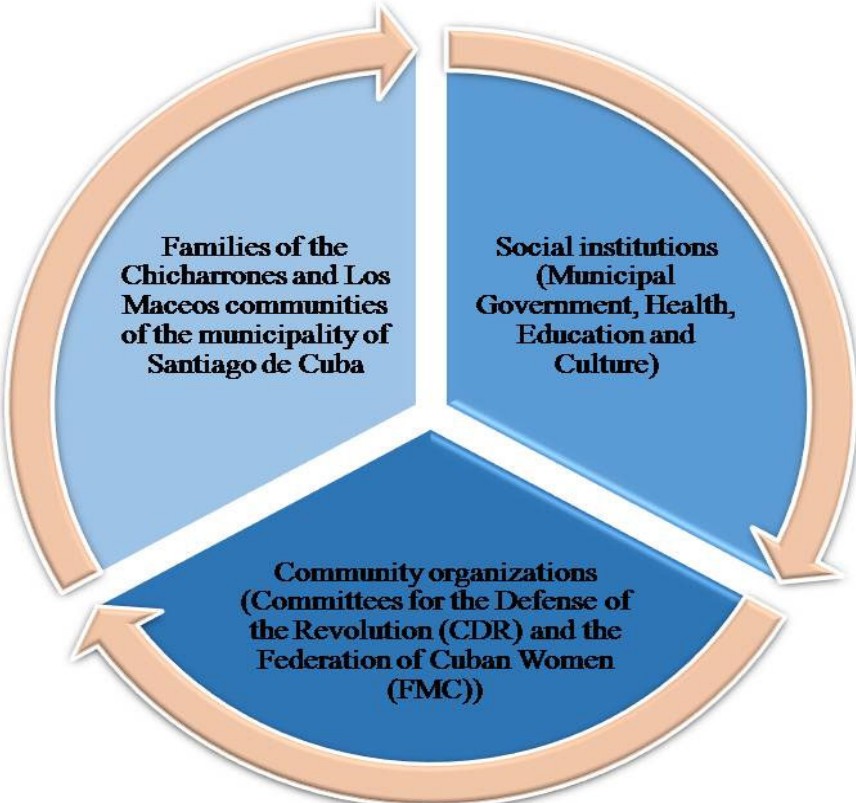

**Figure 7.** Interrelation of the social actors in the social construction of HFS as a complex system. Source: surveys (2021) and authors' own research.

The interplay of these social actors is expressed in the integration of their collective efforts through the greater coordination of behaviors, needs, meanings and experiences in order to achieve common objectives linked to the development of HFS in a pandemic. This proposal sustains that the actors (families, social institutions and community organizations) bring their collective action to fruition in integrated practices of cooperation, where they all dialogue and mutually complement each other in the food space.

To energize this HFS at the community level, families need to adopt a fundamental role based on the food culture in which norms, values, beliefs and practices, constantly produced or replicated, converge. However, the formation of this food culture depends not only on the household framework, but also on other collaborative or social support networks with important roles in the resolution of the food conflicts that have arisen from the current health crisis. Therefore, a system of actions is proposed in which the different social actors participate as collaborative networks of collective action to address the problems associated with the dimensions of HFS in the communities studied (Figure 8).

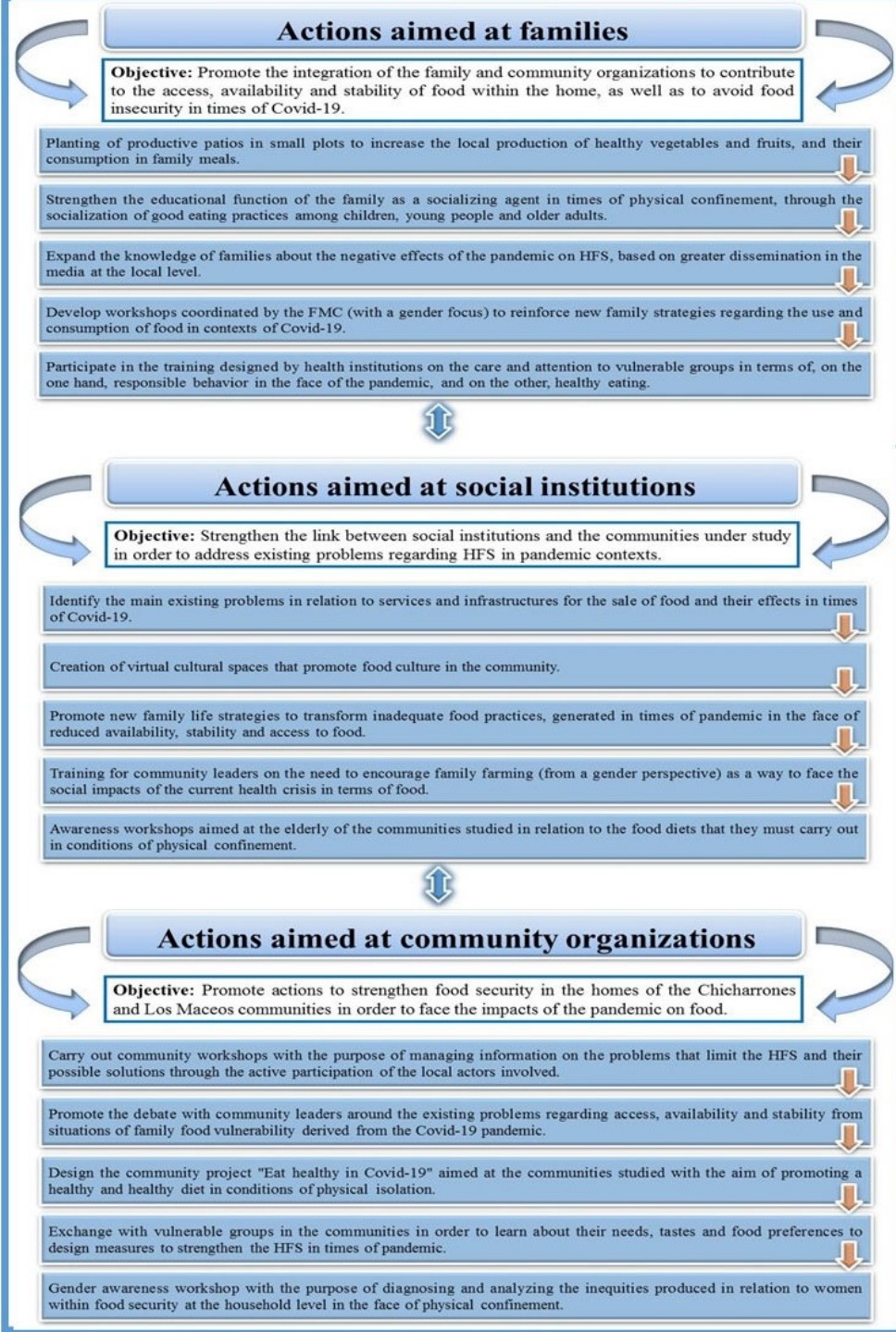

**Figure 8.** Action plan to strengthen the state of HFS.

The fulfilment of these actions requires the active participation of all individual and collective actors, in conjunction with the different factors (social, political, economic, health and cultural), with an impact on the transformation of HFS. Their integration must involve building collective knowledge with new ways of thinking about safe food access and adequate food use at the household level to reduce the impacts of COVID-19. All of this will depend on the commitment of public policies to food distribution, and to bringing it in line with the needs of families, enhancing the resources of both communities.

In order to achieve the sustainability and effectiveness of social policies aimed at food distribution, an effective and participatory evaluation process is required, the social impact of which can transform the way of life of families.

Assessment is the last part of the action plan, and it identifies the shortcomings in the implementation of the action plan, taking into account the joint agreements of all the actors involved in shaping HFS at the community level. The main element of this evaluation process is feedback, with the aim of measuring the impact of the factors (social, economic, political, health and cultural) associated with the pandemic in the family context in order to correct the actions designed and achieve more effective results.

## 6. Conclusions

The theoretical and empirical research carried out on the social construction of household food security as a complex system in these communities in the municipality of Santiago de Cuba led to various conclusions.

Household food security constitutes an important space of exchange between different actors (individual and collective) and factors (social, economic, political and cultural) that intervene in the processes that concern food access, availability, stability and use. Although the social sciences have addressed the study of food problems from various theoretical approaches, the introduction of a complex perspective to handle these interrelationships and understand the changes experienced by families in terms of food in the face of COVID-19 is still insufficient.

In the case of Cuba, the social sciences assume macro analytical positions with regard to food security, highlighting that studies have been reduced to the practices, interactions and meanings constructed regarding food use and consumption at the household level. This shows the need to add new theoretical and methodological approaches to the complex web of relationships, conflicts, communications, vulnerabilities and meanings that HFS socially constructs.

This concept emerges from the current pandemic crisis, with strong impacts on food production, distribution and marketing. Therefore, household food security takes on the condition of a complex system that involves not only actors and factors, but also diverse configurations that are associated with gender, food access, utilization, stability and availability.

The methods and techniques applied reveal that these configurations have materialized in a practical sense: (1) the planning and control capacity of households to cope with the impacts of COVID-19 in the face of instabilities in market prices and available services; (2) insufficient assets and resources available to families to remedy these instabilities, placing them in situations of food vulnerability; (3) new practices associated with food use and consumption in the face of generational dynamics during lockdowns; (4) difficulties in accessing a sufficient amount of food due to the effects of the pandemic; (5) the widening of gender gaps in households due to unequal roles in the preparation and distribution of food, which is accentuated during lockdowns.

Finally, this research has sought to strengthen local food systems by fostering the participation and inclusion of the family network in the construction of food security through good food practices among consumers, taking into account individual and collective sociocultural factors. In addition, this topic line contributes towards the Food Sovereignty and Nutrition Education Plan of the Republic of Cuba (PLAN SAN), which is a legal instrument that extends to all of Cuba. As a theoretical and methodological contribution, it provides an opportunity to recognize the diverse configurations produced in both the urban and rural

communities in the country in terms of food access, use, stability and availability. It also identifies vulnerabilities that affect local food culture and hinder the pursuit of sustainable food sovereignty.

**Author Contributions:** Writing–review & editing, Y.D.R., O.S.N., J.M.J.A. and J.A.M.D. All authors have contributed equally to this paper. All authors contributed to the writing and revising of the manuscript. All authors have read and agreed to the published version of the manuscript.

**Funding:** This publication was made with the financing and support of the CIPHCN.

**Conflicts of Interest:** The authors declare that there are no potential conflicts of interest regarding the research, authorship and/or publication of this article.

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
