# Peer review of "Household Food Security as a Complex System—Contributions to the Social Sciences from the Cuban Perspective during a Pandemic"

_sustainability, doi:10.3390/su141811783_

Round 1

Reviewer 1 Report

To AUTHORS

I found this paper as very interesting and well done.

It is an interesting publication from Caraibean region dealing with households food security in two Cuban province in time of COVID-19 pandemic.  Poverty of households increased by approximately 4 % in 2019. In turn dimension of food security and economical, social factors  decreases. These events inspired the authors to look at HFS in order to modify it and improve the relationship: institutions - organizations - households (Fig. 7), especially during a pandemic.

The work is experimental in nature. In the first phase (preliminary diagnosis of the main social problems 100 questionnaires were collected. In the second, 20 interviews were conducted with families, and 14 in-depth interviews with officials, politicians, public trust persons, 3 with experts and 200 with families. How they were drawn? The latter sample represented about 1% of the population in both communes. What were the sampling, data collection and data processing technique? Please add some information before  print.

On this basis, "Configurations that emerge from HFS as a complex system in times of COVID-19" and 5 conclusions were developed. Based on this, the authors presented an Action plan to strengthen the state of HFS. The results obtained are a contribution to the implementation of the Plan for Food Sovereignty and Nutrition Education of the Republic of Cuba (PLAN SAN), a legal instrument that covers all of Cuba.

Author Response

  • All experts were selected based on their expertise on the subject both nationally and internationally.
  • 14 interviews were conducted with key informants, selected based on intentional opinion sampling (not probabilistic) whose fundamental criterion was the level of knowledge that these local actors have about the constituent elements of family food security in the communities studied.
  • The procedure was carried out based on a sampling error of 10% and a confidence level of 95.57%. This choice was made based on the characteristics of probabilistic sampling, especially random sampling, where its fundamental condition is expressed in the idea that all individuals in the population have the same probability of being chosen in the sample.
  • Likewise, the improvements made on the basis of your comments and suggestions (very valid for the improvement of our contribution) have once again been translated by a translation specialist.

Reviewer 2 Report

Topic of manuscript corresponds to journal area. 

Not clear what is the meaning of Figure 1

Figure 2. Add necessary information related to axis titles

Figure 3, The method of data presentation not siutable.

It is recommended to correct all figures by making data more clearly described

discussions section should be rewritten by adding comparision with data from other researchers works

Author Response

  • Figure 1 expresses the geographical location of the studied communities (Chicharrones and Los Maceos) in the municipality of Santiago de Cuba.
  • The figures clearly show the age ranges and the distribution by sex of the people surveyed from both communities.
  • The figures describe the behavior of the variables (gender, food social vulnerability and practices) in family food security at the level of the communities studied.
  • Likewise, the improvements made on the basis of your comments and suggestions (very valid for the improvement of our contribution) have once again been translated by a translation specialist.

Reviewer 3 Report

Household food security is a meanful topic, especially during this special time under the pandemic.  However, the manuscript were not well orgnized. Judge from the manuscript, 100 or 200 families were surveyed during the study (Line 346 and 369), which makes it confusing. Based on a 200-household-survey, good econometric analysis can done while it looks like the authors didn't dig the data at all.

Author Response

  • It was rectified in lines 354 and 388: there are 200 surveys.
  • It is highlighted that the improvements made based on your comments and suggestions (very valid for the improvement of our contribution) have been translated by a specialist.
  • Likewise, the improvements made on the basis of your comments and suggestions (very valid for the improvement of our contribution) have once again been translated by a translation specialist.

Round 2

Reviewer 2 Report

authors made some corrections, however discussions of results still miss comparision with findings of other researchers 

Author Response

  • In the discussions of the results obtained, a detailed interpretation of the configurations generated around family food security as a complex system is evident, all based on the research background cited in sub-sections 2.1 and 2.2. However, at the request of Reviewer 2, a brief analysis is made pointing out the theoretical guidelines of these authors used in our research, as well as the importance of strengthening these conceptions from deepening the analysis of the configurations that emerge in this system, becoming a superior link in theoretical matter.
  • The improvements made on the basis of the suggestion made by reviewer 2 (very valid for the improvement of our contribution) have gone through the translation of a specialist.
  • The added text is located on page 14, 1st and 2nd paragraph.
  • Thank you